# Characteristics of Strata Behavior and Differentiated Control of Fully Mechanized Mining Working Face with Abnormal Roof

Qiang Fu [1,2,3,4], Ke Yang [1,2,3,4], Xiang He [2,3,4,*], Zhen Wei [1,2,3,4] and Qinggan Yang [1,3,4]

1 State Key Laboratory of Mining Response and Disaster Prevention and Control in Deep Coal Mines, Anhui University of Science and Technology, Huainan 232001, China
2 Institute of Energy, Hefei Comprehensive National Science Center, Hefei 230031, China
3 School of Mining Engineering, Anhui University of Science and Technology, Huainan 232001, China
4 Coal Mine Safety Mining Equipment Innovation Center of Anhui Province, Anhui University of Science and Technology, Huainan 232001, China
* Correspondence: xianghe_cumtb@126.com

**Abstract:** The roof control of a fully mechanized mining face has a significant impact on coal extraction. Therefore, information about strata behavior and differentiated control measures for abnormal roofs are needed. In the present research, we used the ground pressure theory to calculate the interval of the first and periodic weighting of working faces 11113, 11213, and 11313 in the Panbei mine. The maximum support working resistance required by the three working faces was calculated considering the influence of the hard roof, fault zone, and skip mining mode on the relationship between support–surrounding rock. In addition, we determined the hydraulic support for the three working faces taking into account the impact of roof lithology on ground pressure, as well as the layout and mining sequence of the working face. In order to ensure the safety of the working face mining, we implemented the laying mesh method, controlling the roof, providing side protection in the fault-affected area, and using the violent ground pressure control method for advanced deep hole pre-split blasting under hard roof conditions. The engineering practice showed that the effective control of roof ground pressure in abnormal areas was achieved using these control measures.

**Keywords:** ground pressure; numerical simulation; hard roof; tectonic influence; pre-split blasting





## 1. Introduction

The geological conditions of coal strata in China are complex and diverse. Even in the same mining area and the same coal seam, roof lithology and rock formation are different [1,2]. Engineering practices have shown that lithology and structural characteristics of coal seam roofs are the key factors affecting the design parameters of fully mechanized mining working face, equipment selection, and ground pressure control [3–5]. In most mining areas, the same set of fully mechanized mining equipment is used in multiple working faces with different roof lithologies. These differences complicate the performance of the equipment [6]. Since soft coal presents a diversity of roof conditions including soft roof and floor and given that thick and hard rock formation may result in faults and direct overload, a significant amount of research has been carried out to determine effective working face mining parameters and roof control designs.

With the aim of solving technical problems related to ground pressure control in "Three Soft" large mining heights, Tu et al. [7] optimized the working face parameters through the surrounding rock control theory and technology as well as a comprehensive gas treatment technology. Bahrani et al. [8] addressed issues associated with the simulation of jointed rock masses and ground support by employing both continuum and discontinuum numerical methods. Li et al. [9] proposed and validated an improved support method for roadways in broken rock mass under high geo-stress. These approaches were the basis for safe mining practices in the "Three Soft" coal seam with large mining height working faces.

Dychkovskyi et al. [10] developed a rendering algorithm of a 3D model of rock mass in terms of long-pillar mining of a coal seam using double-unit longwalls. Wang et al. [11] analyzed the mechanisms of spalling in very soft coal seam walls and proposed that the main technical ways to prevent and reduce the spalling of very soft coal seam walls were to reduce coal pressure and improve the shear strength of coal. Later, Vu proposed solutions to prevent face spalling and roof falling in fully mechanized longwalls in underground mines in Vietnam [12]. The effect of mining's advancing direction on the stability and control of the surrounding rock in the fault zone of the working face has also been studied. Babets et al. [13] developed a technique for rock strength assessment considering the rock mass disintegration and watering in the fault area. Li et al. [14] studied the mining stress in rock mass at the fault zone. These researchers used similar material simulation tests and determined the influence of the fault zone on stress transmission. Their results provided the basis and reference for reasonable reservation of the fault coal pillar and roadway layout. Additionally, the hinge balance theory of the main roof fault rock block was used by Li et al. [15] to study the load transmission law and difference of the main roof of work face goaf on both sides of the fault. Dai et al. [16] determined the first mining hanging wall and the first mining footwall of the fault and discussed the relationship between the retention of the fault coal pillar and the advancing direction. Ju et al. [17] focused on large overhang spans, severe weighting, and difficulties in controlling thick and hard roof working faces. These researchers applied the key layer theory to determine the mechanism of interaction between key layer breaking and ground pressure. Fan et al. [18,19] determined disaster characteristics and proposed technical measures for preventing hard roof impact ground pressure. Cui et al. [20] investigated the load distribution, energy accumulation, and release, as well as the weighting strength characteristics of the first and periodic weighting in the process of hard roof fracture. Guo et al. [21] discussed the technique of controlled caving of hard roofs and proposed a method to calculate the reasonable suspended roof length. He et al. [22,23] used numerical simulation to analyze the mechanism of deep hole pre-splitting blasting and hydraulic pre-splitting roof weakening. Sun et al. [24] proposed a model based on the traditional CCM and the classical Carranza's model to analyze support-surrounding rock interaction considering pre-reinforcement. He et al. [25–27] applied deep hole pre-splitting blasting and hydraulic directional fracturing method to perform industrial experiments on hard roof weakening with good results. Yang et al. [28] analyzed the relationship between support-surrounding rock under hard roofs of thin bedrock conditions. These researchers calculated the support working resistance, and proposed methods for controlling fully mechanized mining working faces. In summary, fully mechanized mining technology has been developed and applied to complex and diverse mining coal seams. However, most of these studies have considered the ground pressure and control of working faces under the same coal seam roof lithology. Little research has been performed to determine the pressure and roof control of adjacent working faces with different coal seam roof lithologies.

Hence, based on experimental analysis and theoretical calculations, we analyzed the characteristics of ground pressure in the fully mechanized mining working faces of the zone with an abnormal roof and the mechanism of roof lithology changes on ground pressure. With this information, we proposed the mechanisms for selecting proper fully mechanized mining equipment and ground pressure control methods. This information is of great significance for safe and efficient mining.

## 2. Case Study

The A3 coal was obtained from group A coal roof in layers of the Panbei Mine in the Huainan mining area, China. The first mining working face was 11113, followed by 11313, and finally working face 11213. The working face parameters are shown in Figure 1. The comprehensive analysis of the borehole columnar diagram and field borehole coring data indicated that, along the direction of the coal seam, the immediate roof thickness gradually decreased and tended to be directly covered by a thick and hard roof, as shown in Figure 2.

In addition, the coal seam angle gradually increased, and the mining area displayed dense faults with a density of 25 faults/km². These issues complicated the design and safe mining of working faces. The main physical and mechanical parameters of rocks and coal of the roof and floor of this mining area were obtained in the laboratory using the coal and rock physical parameters test.

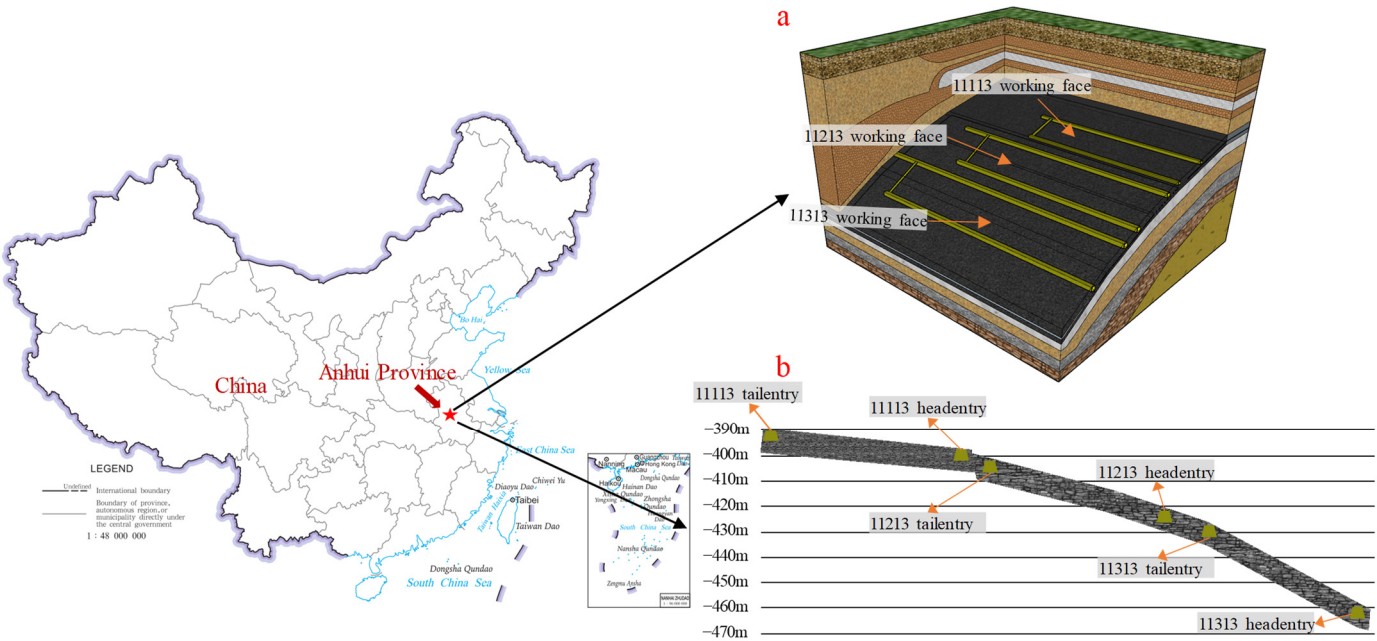

**Figure 1.** Layout and section of mining area. (**a**) Plane layout of mining area, working face of group A coal. (**b**) Section of group A coal, mining area along the uphill direction.

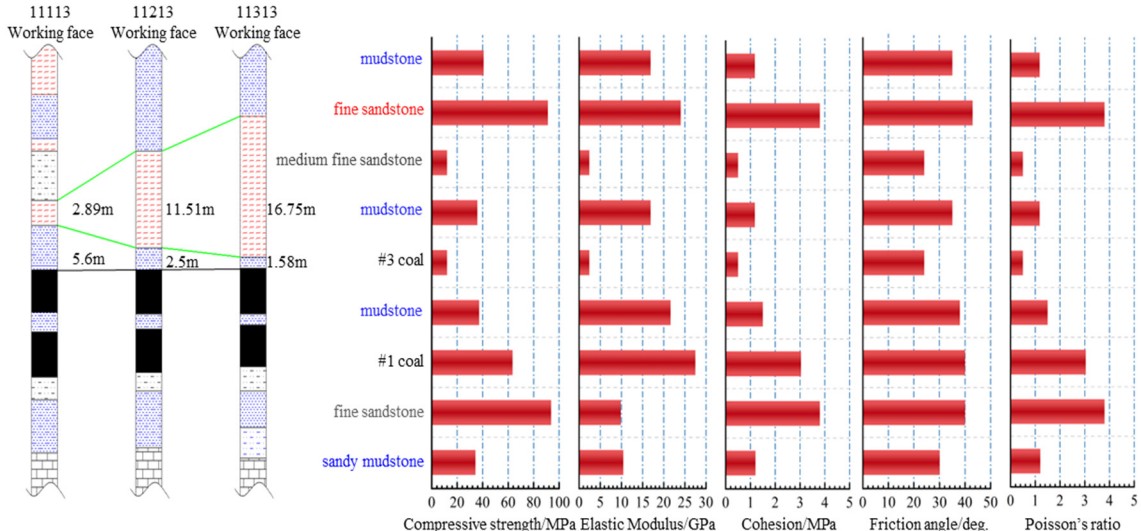

**Figure 2.** Comprehensive histogram and physical and mechanical parameters of different working faces.

According to the main roof classification method, the $N$ ratio of immediate roof thickness $h_i$ to coal seam mining height $h_m$ is expressed as $N = h_i/h_m$. Description of the three basic types of working face roof rock formations were:

1.  Working face 11113, $N > 5$, belonged to class I roof condition. The thicknesses of overlying rock formations were small, and the roof rock contained fractures.
2.  Working face 11213, $2 < N < 5$, belonged to class II roof condition. There was an immediate roof of size 2.0~9.0 m between the main roof and the coal seam.

3. Working face 11313, *N* < 2, belonged to class III roof condition. The lithology of the main roof is medium-fine sandstone. A small amount of 1~2 m mudstone roof was contained between the main roof and the coal seam and almost no immediate roof was present.

## 3. Characteristics of Strata Behavior with Abnormal Roof

### 3.1. Caving Characteristics of Abnormal Roof

In the present research, we considered the engineering conditions and used the calculation method of periodic weighting interval previously reported in [11,14]. The following assumptions were made: in the overlying load of the main roof calculation, the rock formation thickness was considered a real value; in the case of stress and fracture interval of rock mass, rock formation thickness was taken as the equivalent rock formation thickness described by the rock quality designation (RQD) index.

The main roof's first weighting interval *L* was calculated using Equation (1):

$$L = h\sqrt{\frac{\sigma_t}{q}} \tag{1}$$

where *L* is the fracture interval when the main roof was calculated as beam (m); *h* corresponds to the thickness of the main roof rock formation (m); $\sigma_t$ represents the main roof rock formation tensile strength (MPa); *q* is the load bear by main roof rock formation (MPa).

The overlying load rock formation q on the main roof [11] was calculated using Equation (2):

$$(q_n)_1 = \frac{E_1 h_1^3 (\gamma_1 h_1 + \gamma_2 h_2 + \cdots + \gamma_n h_n)}{E_1 h_1^3 + E_2 h_2^3 + \cdots + E_3 h_n^3} \tag{2}$$

where $(q_n)_1$ represents the load acting on the main roof rock formation in the nth layer (kPa); $E_1$, $h_1$ are the elastic modulus (GPa) and thickness (m) of main roof rock formation; $E_n$, $h_n$ indicate the elastic modulus (GPa) and thickness (m) of rock formations above the main roof; $\gamma_1$, $\gamma_n$ correspond to bulk density of the main roof and above the rock formation (kN/m$^3$).

When $(q_{n+1})_1 < (q_n)_1$, q in the first weighting interval becomes $(q_n)_1$.

According to the coring test, the RQD value of medium-fine sandstone in the main roof was 0.60. The main roof's first weighting interval and periodic weighting interval were calculated for the three working faces considering the mechanical parameters of the rock in the borehole coring. Results are shown in Table 1.

**Table 1.** Main roof weighting intervals of different types of roofs.

| Working Face | Roof Category | $q_1$ (kPa) | $(q_2)_1$ (kPa) | $(q_3)_1$ (kPa) | First Weighting Interval (L/m) | Periodic Weighting Interval (m) |
|---|---|---|---|---|---|---|
| Working face 11113 | I | 74.0 | 33.6 | / | 29.6 | 12.1 |
| Working face 11213 | II | 309.2 | 417.4 | 95.2 | 38.0 | 15.5 |
| Working face 11313 | III | 449.6 | 638.7 | 108.7 | 44.7 | 18.2 |

### 3.2. Weighting Characteristic of Abnormal Roof

3.2.1. Working Face 11113

In the working face 11113 (Class I roof), no main roof rock formation was observed in the collapse zone. It is likely that the immediate roof rock formation fell during the mining of the working face, and that the covered 4.2 m-thick fine sandstone (20 m from the coal seam) entered the fracture zone. Thus, hydraulic support carried the load of rock mass in

the collapse zone within the roof control zone, as shown in Figure 3. The first weighting of the main roof was not significant.

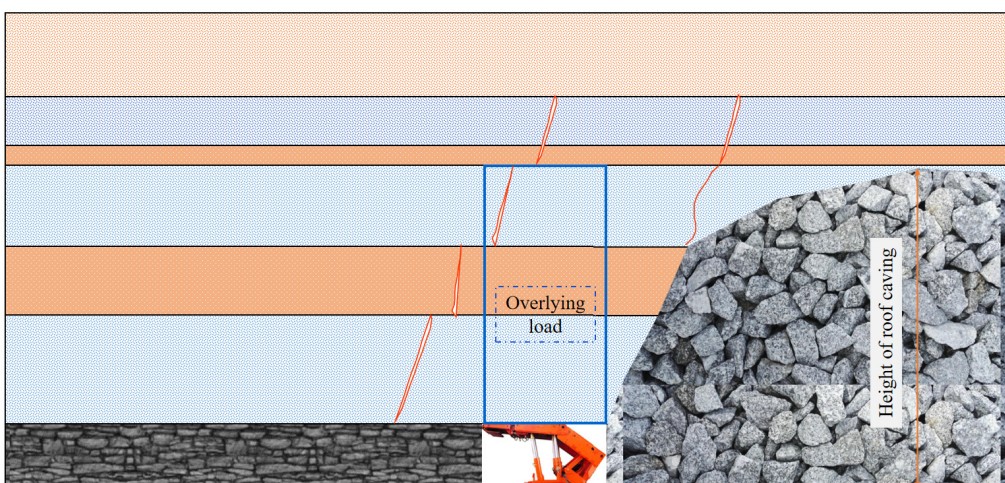

**Figure 3.** Loading on hydraulic support in the longwall panel of working face 11113.

The support stress was calculated using Equation (3) [11]:

$$P = \frac{L_a \cdot \gamma h L_z' \cdot \cos\alpha \cdot K}{\eta'} \tag{3}$$

where $L_a$ indicates the length of the working face controlled by each support (m), which was set as 1.75 m; $\gamma$ corresponds to the bulk density of immediate roof rock formation in the collapse zone (kN/m$^3$), 25 kN/m$^3$; $h$ is the immediate roof thickness in the collapse zone (m), 9~15 m; $L_z'$ represents the immediate roof length of rock beam (m), $L_z' = L_d + L_h + L_z'$ ($L_d$ is end distance; $L_h$ is the sum of top beam and front beam length (m); $L_d + L_h$ indicates the maximum top control distance (m); $L_{zx}$ represents the limit overhang length of immediate roof rock formation after support (m), $L_z' = 6 + 1 = 7.0$ m; $\alpha$-coal seam dip angle (°), 10; $K$ is the safety factor, 1.2; $\eta'$ is the coefficient considering the waste dump on the shield beam and that the top beam was not perpendicular to the column, 0.85.

The support working resistance required for working face 11113 was 3832~6386 kN.

### 3.2.2. Working Face 11213

According to the sequence of the working face 11113-11313-11213, working face 11213 corresponded to an island working face. F11213-1 fault exposed at 31 and 43 m away from the setup entry of the working face in tailentry and headentry, respectively, also affected the first weighting of the working face. The mining influence and overlying strata are shown in Figure 4.

The two-layer hard rock formations in the roof of working face 11213 were located within the elevation of the goaf fracture zone (45 m). The roof of working face 11213 was affected by the mining of working faces 11113 and 11313 and their corresponding faults. For this reason, a large fracture was observed as a result of the damage in the inclined direction. As shown in Figure 4, working face mining may cause gradual collapse along the slope, forming a simple support on three sides and a fixed support on one side. The support stress was calculated using Equation (4) [11], which considered the periodic fracture of the main roof.

$$P = \frac{L_a\left(\gamma h L_z' + \sum_{i=1}^{n} \gamma_{li} h_{li} L_{lki}\right)\cos\alpha \cdot K}{\eta'} \tag{4}$$

where $\gamma_{l1}$ corresponds to the average bulk density of the first main roof and its additional rock formation in the collapse zone (kN/m³), 26 kN/m³; $h_{l1}$ is the thickness of the first main roof and its additional rock formation in the collapse zone (m), 15 m; $L_{lk1}$ indicates the rock block length of the first main roof in the collapse zone (m), 7.7 m (half of the main roof periodic weighting interval); values of other parameters were: $L_a$ = 1.75 m, $\gamma$ = 25 kN/m³, $h$ = 2.5 m, $L'_z$ = 7.0 m, $\alpha$ = 18°, $K$ = 1.2, $\eta'$ = 0.85.

Later, we substituted the corresponding data into Equation (4). Our calculations indicated that the working resistance required to support working face 11213 was 8084 kN.

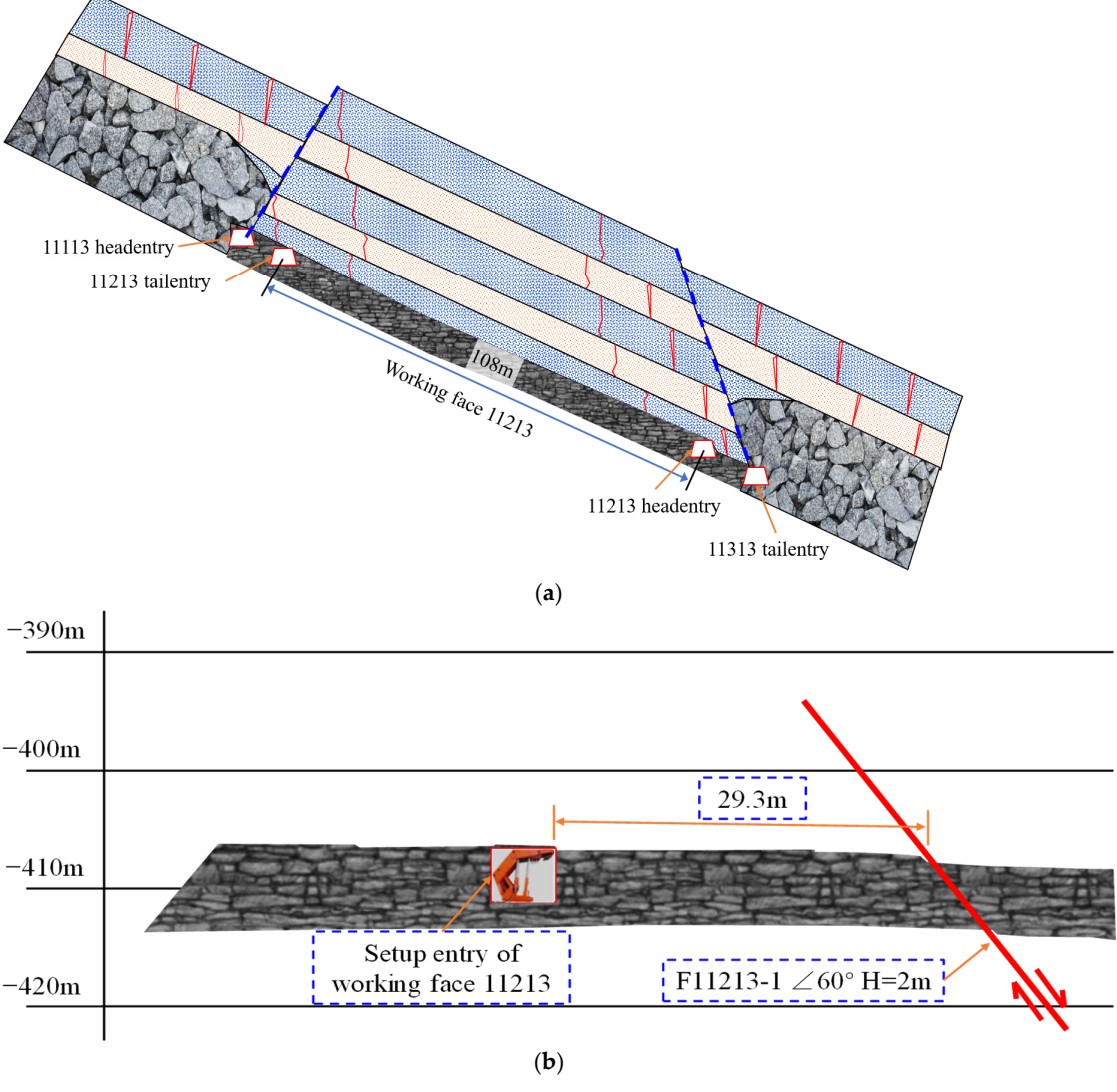

**Figure 4.** Structure of overlying strata near the setup entry in the longwall panel of working face 11213. (**a**) Rock structure in the inclined direction. (**b**) Fault influence (along the tailentry section).

### 3.2.3. Working Face 11313

Working face 11313 corresponds to class (III) with a thin immediate roof where the local main roof directly covered the working face. As shown in Figure 5, according to the section along the headentry direction near the setup entry, the lower part of the working face was affected by FX1 and F11313-9 faults. The tailentry and headentry exposed F11313-9 faults at 110 m and 26.3 m away from the setup entry, respectively. The upper part of the working face was not affected by the fault within the first weighting range.

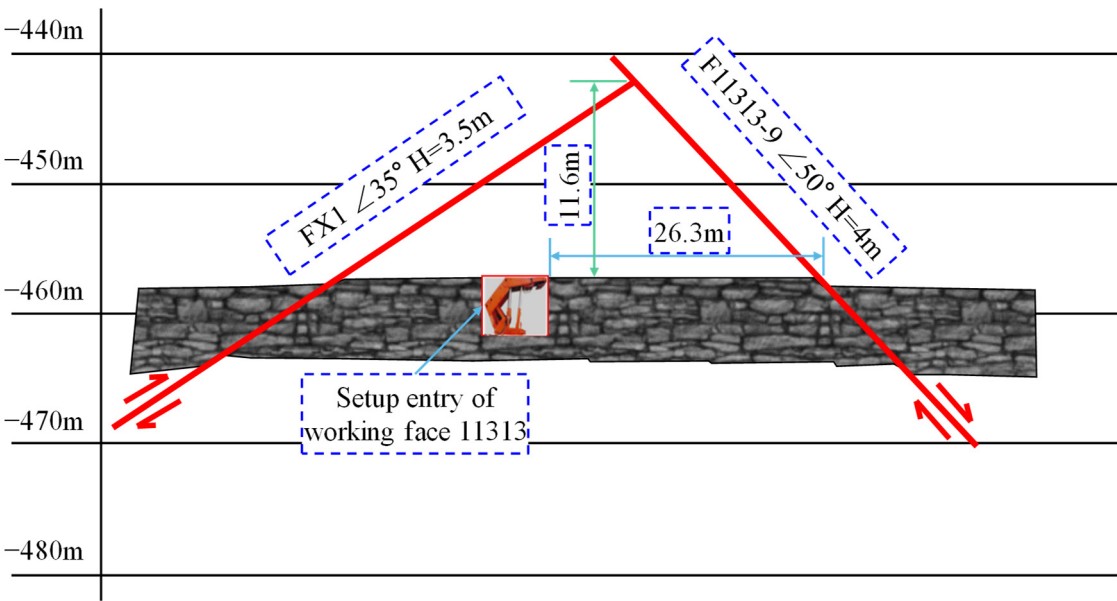

**Figure 5.** Section along the longwall panel near the setup entry in working face 11313.

(1) Assuming that the first layer of the 16.75 m thick medium-fine sandstone did not separate during the collapse of the whole layer, the 8.3 m thick sandy mudstone, which displayed certain stability, entered the fracture zone. In addition, the first weighting interval was 44.7 m. According to Equation (4), the support working resistance required for the first weighting main roof was as high as 239 kN.

(2) The on-site coring near the setup entry of the working face indicated that the rock formation in the roof developed fissures and presented poor integrity. Data indicated that the average uniaxial compressive strength of medium-fine sandstone was relatively low, with a value of 63.5 MPa and a statistical average RQD of 0.60. According to this, the product of the actual thickness of medium-fine sandstone and RQD value (defined as equivalent thickness in this case) was 16.75 × 0.6 = 10 m. Thus, the support working resistance required for working face 11313 for the first weighting was 14,132 kN.

In summary, the hydraulic support working resistance required during the first weighting of the three working faces varied greatly (3832~14,132 kN). This probably occurred because of differences in roof lithology in the mining area. However, the support working resistance required for periodic weighting was 6135~9842 kN, which affected the unified selection of mining support and fully mechanized mining equipment.

## 4. Numerical Simulation

### 4.1. Model Establishment

According to the geological characteristics of the working face 11313, a numerical simulation model of the thick hard roof was obtained using FlAC3D. The size of the model was selected as length × width × height = 250 m × 250 m × 200 m. In addition, the elastic-plastic constitutive was selected, the Mohr–Coulomb criterion was used for the failure criterion, and the goaf was filled with weakened material. Displacement boundaries were used at the bottom and on both sides of the model, and stress boundaries are used at the upper part of the model, As shown in Figure 6. In order to simulate the action of the overlying strata, an equivalent uniform load of 9.3 MPa was applied to the top of the model, the lateral pressure coefficient was 1.16, and the horizontal stress of the model was 10.8 MPa. According to Zhao et al. [29–31], the impact factor D of blasting excavation is 0.7 and the details are shown in Table 2.

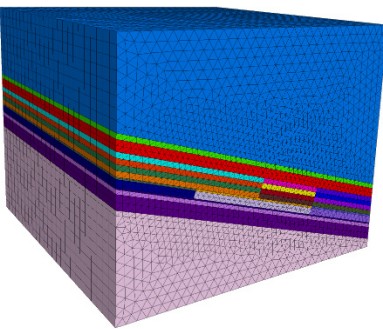

**Figure 6.** Numerical model of 11313 working face.

**Table 2.** Value of damage factor after pre-split blasting.

| Rock Mass Characteristics | Damage Factor D |
|---|---|
| The blasting effect is poor, and the hard rock roadway is partially damaged | D = 0.8 |
| Small and medium-scale blasting caused blasting of rock mass | D = 0.7 Good blasting effect D = 1.0 Poor blasting effect |
| Mass production blasting caused significant disturbances | D = 1.0 Good blasting effect D = 0.7 Poor blasting effect |

The relationship between the deformation parameters of the rock mass before and after the damage is shown in Equation (5). The strength parameters, cohesion, and internal friction angle after blasting were 0.5 times [32,33] those before blasting. The basic physical and mechanical parameters of the roof after blasting are shown in Table 3.

$$\begin{aligned} E_1 &= (1 - D)E \\ \mu_1 &= \mu \end{aligned} \tag{5}$$

where $E$, $G$, $\mu$ are the elastic modulus, shear modulus, and Poisson's ratio of the undamaged rock, respectively; $E_1$, $G_1$, $\mu_1$ are the corresponding quantities of rock blasting damage, respectively.

**Table 3.** Physical and mechanical parameters of the basic roof after blasting.

| Lithology | Bulk Modulus/Gpa | Shear Modulus/GPa | Cohesion/MPa | Internal Friction Angle/° |
|---|---|---|---|---|
| Medium-fine sandstone | 3.6 | 3.9 | 1.5 | 20 |

*4.2. Analysis of Results*

The stress of the surrounding rock in the stope is shown in Figure 7. The stress arch rotates to the blasting side, and the stress arch height increases with the blasting depth, from spoon shape to oval shape. When the blasting thickness is less than 9 m, the stress arch height increases gradually from 30 m to 40.2 m with the increase in blasting thickness. The inclination angle of the stress arch remained unchanged, the side inclination angle of the working face was 60°, and the inclination angle of the stress arch on the lower side of the working face was about 20°. When the blasting thickness was more than 12 m, the stress arch height increased to 54.2 m, the inclination angle of the stress arch on the side of the working face was 60°, and the inclination angle of the stress arch on the lower side of the working face suddenly changed to 57°. The side abutment pressure of the headentry was significantly released, which benefits the support.

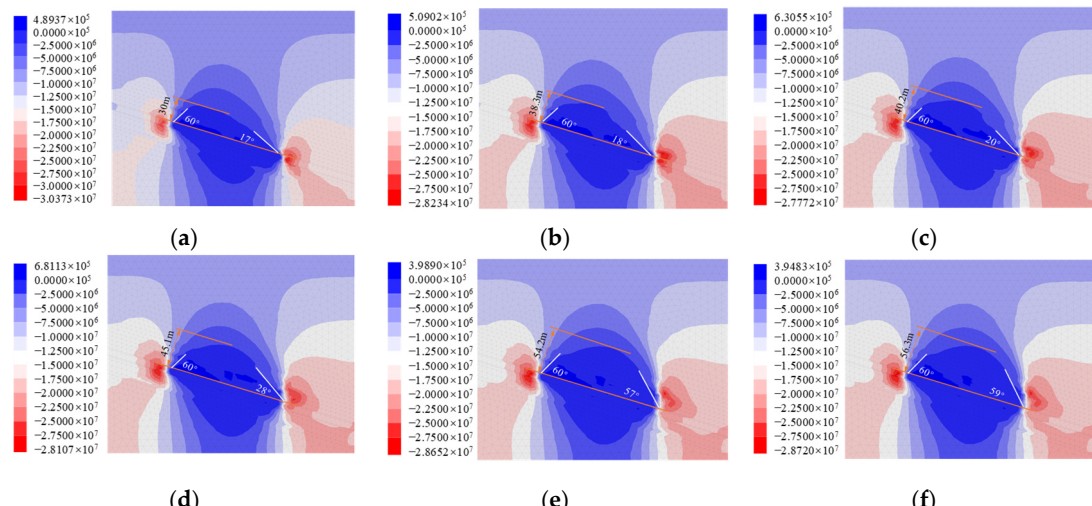

**Figure 7.** Stress of surrounding rock along working face tendency after blasting. (**a**) The thickness of the main roof pre-split blasting is 0 m; (**b**) The thickness of the main roof pre-split blasting is 3 m; (**c**) The thickness of the main roof pre-split blasting is 6 m; (**d**) The thickness of the main roof pre-split blasting is 9 m; (**e**) The thickness of the main roof pre-split blasting is 12 m; (**f**) The thickness of the main roof pre-split blasting is 15 m.

Figure 8 shows the distribution of the abutment pressure when the thickness of the main roof pre-split blasting was 12 m. In the range beyond 80 m in front of the coal wall, the front abutment pressure also remained constant, and the coal body entered a state of in-situ stress. In the range of 25~80 m in front of the coal wall, the coal body was affected by the mining disturbance. As the distance from the coal wall decreased, the abutment pressure above the coal body also gradually increased. When the distance from the coal wall was 20 m~25 m, the abutment pressure reached a peak value of 23.4 MPa, and the stress concentration factor reached 1.67 (the in-situ stress is 13.7 MPa). Compared with the tailentry, the influence of the front abutment pressure on the blasting zone and the distance of the peak abutment pressure from the working face increased. The peak value of the abutment pressure also decreased to 16.2 MPa, and the stress concentration factor was reduced to 1.3. The degree of stress concentration on the coal body in front of the working face can be significantly reduced after the roof blasting, and the stress environment can be effectively improved.

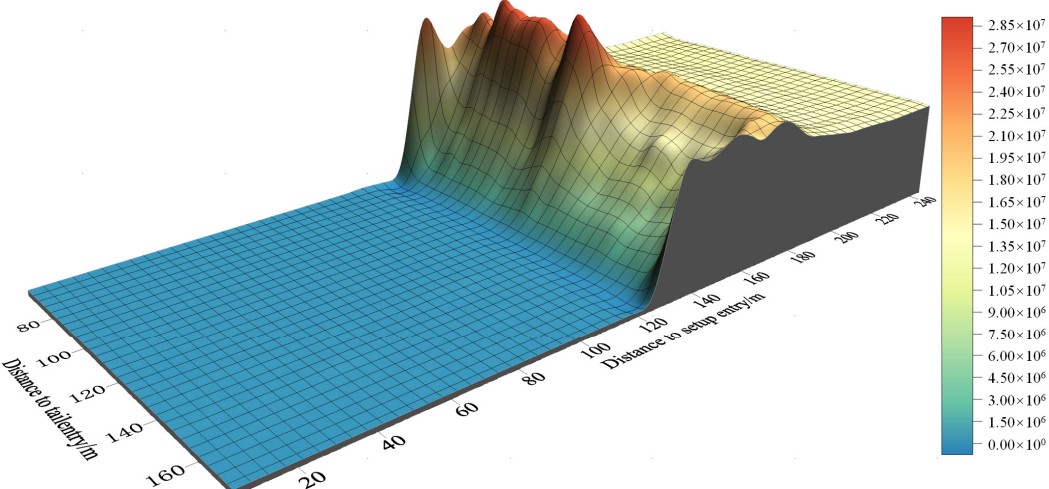

**Figure 8.** Distribution of abutment pressure.

## 5. Ground Pressure Control

### 5.1. Support Selection of Working Face

Since working face 11113 was the first mining face of A3 coal in the mine, and considering the previous experience of the Huainan mining area with fully mechanized mining, it was proposed that mining should be performed in two layers, with an average mining height of the top layer of 2.8 m. With respect to working face 11313, the theoretical analysis showed that the rated working resistance of 9200 kN support did not satisfy the strength requirements of first weighting, and the support working resistance required for working face periodic weighting was also in danger of exceeding the rated working resistance. In order to unify mining area technologies and equipment, and to reduce the production cost of the working face, the deep hole pre-splitting roof technology was selected in order to reduce the strength of working face 11313 during the first weighting and local regional periodic weighting. Thus, after performing theoretical calculations for working face 11113 ground pressure, the hydraulic support models of the three working faces were ZZP7200/18/38, ZZ9200/24/50, and ZZ9200/24/50, respectively.

### 5.2. Scheme of Deep Hole Pre-Split Blasting

5.2.1. Initial Caving

The lower roof of working face 11313 was affected by the fault, and the integrity of the main roof was damaged. Hence, a hole in working face 11313 5 m away from the setup entry was drilled along the slot and directed to the roof. In addition, blasting was carried out. The blast hole layout scheme is shown in Figure 9, and the blasting technical parameters are shown in Table 4.

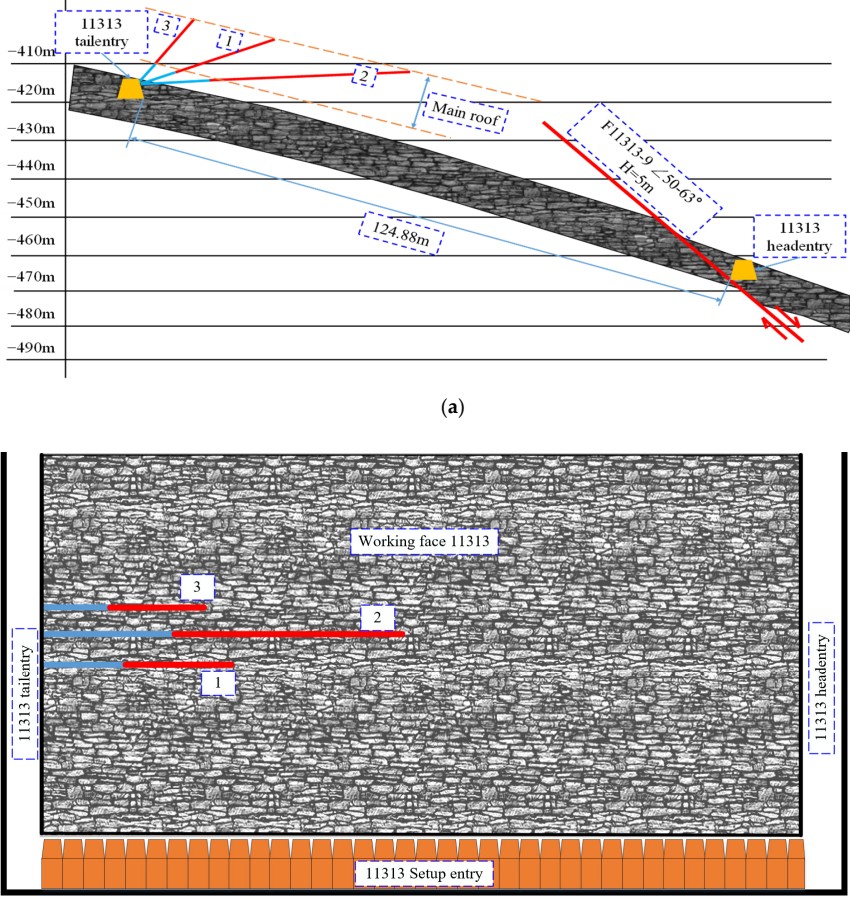

(a)

(b)

**Figure 9.** Layout of roof blast hole near 11313 setup entry. (**a**) Blast hole layout in the inclination direction; (**b**) Blast hole layout in the strike direction.

**Table 4.** Technical parameters of deep hole blasting near setup entry.

| Roadway Name | Blast Hole Serial Number | Blast Hole Depth/m | Dip Angle/° | Blast Hole Diameter/mm | Charge Length/m | Blocking Length/m |
|---|---|---|---|---|---|---|
| 11313 tailentry | 1# | 19 | 34 | 94 | 14 | 5 |
| | 2# | 55 | 3 | 94 | 45 | 10 |
| | 3# | 17 | 70 | 94 | 16 | 4 |

### 5.2.2. Periodic Weighting Caving

In some areas of working face 11313, the main roof thickness was greater than 16.75 m. It was also observed that the main roof directly covered the working face, which displayed good integrity, with the roof hanging over 15 m. The advanced deep hole blasting technology was applied to these areas with frame pressing risk, as shown in Figure 10. According to the engineering geological analysis, the deep hole blasting presplitting roof technology should be selected when the fault structure is in use, details are shown in Table 5.

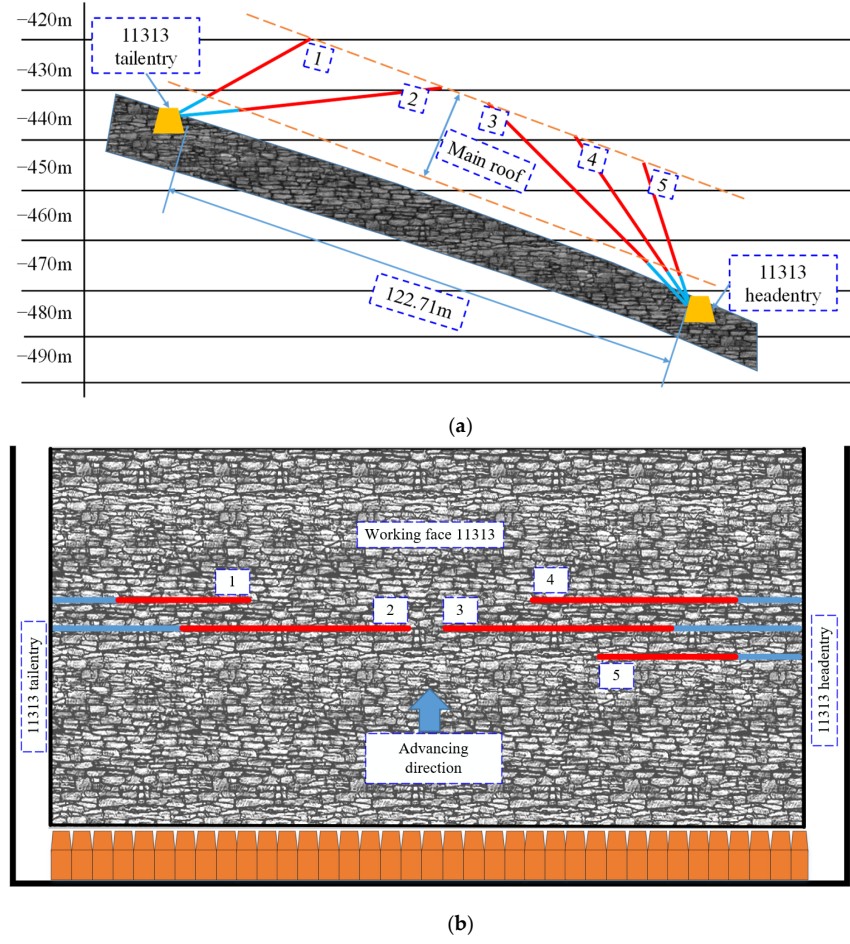

(a)

(b)

**Figure 10.** Layout of roof blast hole during advancing mining of working face 11313. (**a**) Blast hole layout horizon. (**b**) Blast hole parameters.

**Table 5.** Technical parameters of long suspended roof section deep hole blasting.

| Roadway Name | Blast Hole Serial Number | Blast Hole Depth/m | Dip Angle/° | Blast Hole Diameter/mm | Charge Length/m | Blocking Length/m |
|---|---|---|---|---|---|---|
| 11313 tailentry | 1# | 27 | 34 | 94 | 21 | 6 |
| | 2# | 65 | 9 | 94 | 55 | 10 |
| 11313 headentry | 3# | 65 | 45 | 94 | 55 | 10 |
| | 4# | 37 | 65 | 94 | 30 | 7 |
| | 5# | 30 | 85 | 94 | 24 | 6 |

*5.3. Roof Control Scheme in Fault Zone*

Since the fault zone on the roof was relatively broken and difficult to control and considering that a layered mesh laying under the layered mining was needed, the mesh laying method was improved. Thus, we proposed the roof mesh laying and hanging method with advanced hanging mesh protection, as shown in Figure 11. This method was useful in controlling the roof fall and rib spalling in the fault zone and broken roof area.

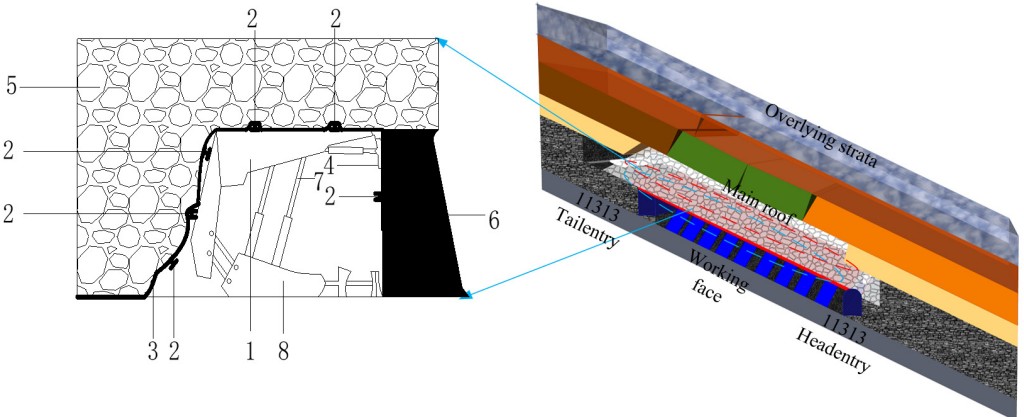

**Figure 11.** Support cross-section of rib spalling and roof fall protection for working face (1-hydraulic support top beam; 2-I steel; 3-anchor chain mesh; 4-side protecting board; 5-workface broken roof; 6-workface coal wall; 7-hydraulic prop; 8-support base).

*5.4. Field Measurement of Ground Pressure*

According to field measurements, the first weighting interval of working face 11113 was 33 m and the weighting strength was 34 MPa. Thus, the support met the mining demand. During weighting, the average dynamic loading efficiency was 1.3, and the weighting was relatively stable. The broken roof mesh laying technology had a good control effect on the roof, the weighting interval basically conformed to the theoretical value, and the hydraulic model met the mining demand.

When mining the 27 m in working face 11213, the main roof broke and collapsed. The working face weighting strength was reduced by increasing the initial force and reducing the mining height. The length of the periodic weighting step was 15 m, which was basically consistent with the theoretical weighting interval, and the weighting strength was within the bearing range of support.

When the deep hole blasting technology was applied to working face 11313, the large weighting interval and weighting strength were avoided. After advancing about 15 m into the main roof rock formation, layer separation and caving occurred. As shown in Figure 12, during periodic weighting, the support value reached the support resistance required by the working face. In the subsequent mining process, the advanced deep hole pre-splitting technology effectively prevented the risk of frame pressing in the working face.

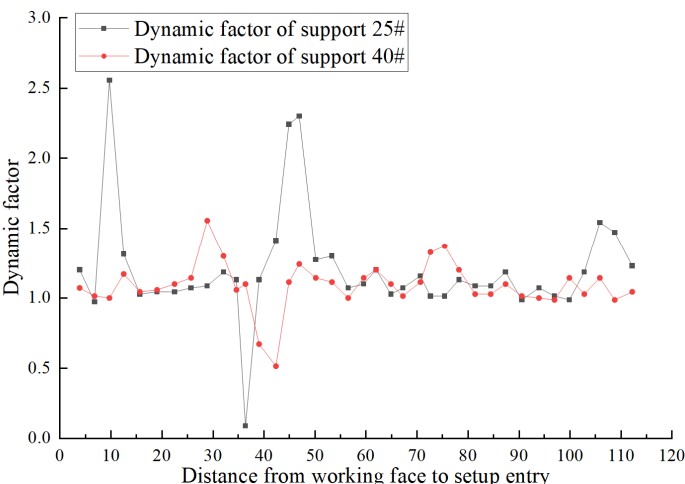

**Figure 12.** Dynamic load coefficient near blasting zone.

## 6. Discussion

Deep hole pre-split blasting has been proven to be an effective technical measure to control the ground pressure within a certain limit [22–25]. An effective method based on deep hole pre-split blasting to control roof collapse for the adjacent working faces with different roof conditions has been proposed in this paper. In order to determine the working face parameters and roof control design with different roof conditions in the same mining area, coal seam characteristics and roof differences were considered. The geo-mechanical evaluation of the surrounding rock and the overall prediction of the ground pressure within the mining area were the basis for the selection of fully mechanized mining equipment and ground pressure control.

First of all, it is necessary to obtain detailed information on the roof strata of adjacent working faces and conduct theoretical calculations to find the hard strata [34]. Secondly, according to the results of theoretical calculations, it is necessary to identify the range of pressure variation that the working face may face during the process of mining. Finally, in the process of mining, the characteristics of roof pressure, numerical simulation, and field monitoring results must be used to develop the design of the weakened roof scheme for different working faces.

Although the method has been successfully applied to the working face 11313, it is still necessary to carry out a detailed study to determine the scope of application and propose a suitable model for the control of the stopping roof of the adjacent working face.

## 7. Conclusions

Based on theoretical analysis, numerical simulation, and field measurements of ground pressure induced by working face 11113, 11213, and 11313 mining, the main conclusions are as follows:

(1) As the deep hole pre-split blasting thickness of the main roof increases, the stress arch rotates to the blasting side, and the stress arch height increases with the blasting depth from spoon shape to oval shape. The stress arch height increased to 54.2 m, with an inclination angle of the stress arch on the side of the working face of 60°.

(2) Herein, a two-deep hole pre-splitting blasting scheme to set up the entry section and long suspended roof section was proposed. This helped reduce the first weighting strength of working face 11313 below 34 Mpa and the roof suspension distance within 15 m, which effectively weakened the impact of severe ground pressure.

(3) The roof mesh laying and hanging method with advanced hanging mesh protection was proposed. This method effectively controlled roof fall and rib spalling.

**Author Contributions:** Conceptualization, Q.F.; Data curation, Z.W.; Formal analysis, X.H.; Investigation, K.Y.; Resources, Q.Y.; Writing—original draft, Q.F.; Writing—review & editing, K.Y., X.H. and Z.W. All authors have read and agreed to the published version of the manuscript.

**Funding:** This study was supported by the Institute of Energy, Hefei Comprehensive National Science Center under Grant No. 21KZS217 and No. 21KZS215, Major special projects of science and technology in Shanxi Province (No. 20191101016), Open Research Grant of Joint National-Local Engineering Research Centre for Safe and Precise Coal Mining (Grant No. EC2021014).

**Data Availability Statement:** The data used for conducting classifications are available from the corresponding author upon request.

**Acknowledgments:** All authors contributed to this paper. Qiang Fu prepared and edited the manuscript. Xiang He, Ke Yang and Zhen Wei substantially contributed to the data analysis and revised the article. Qinggan Yang reviewed the manuscript and processed the investigation during the research process. And we would like to express our thanks to Yanqun Yang from Shanxi Coking Coal Group Co., Ltd. for his support and assistance with the manuscript. This study was supported by the Institute of Energy, Hefei Comprehensive National Science Center under Grant No. 21KZS217 and No. 21KZS215, Major special projects of science and technology in Shanxi Province (No. 20191101016), Open Research Grant of Joint National-Local Engineering Research Centre for Safe and Precise Coal Mining (Grant No. EC2021014).

**Conflicts of Interest:** The authors declared no potential conflict of interest concerning this article's research, authorship, and/or publication.

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
