# Peer review of "Characteristics of Strata Behavior and Differentiated Control of Fully Mechanized Mining Working Face with Abnormal Roof"

_sustainability, doi:10.3390/su142013354_

Round 1

Reviewer 1 Report

This article is brilliantly written, and the presentation of problems and answers is really impressive and easy to follow. Theories that were relatively new were highlighted and put to use. The application of ground pressure theory to compute the initial and periodic weighting intervals of working faces. It appears to be more efficient and useful, and it is the appropriate theory to apply in this scenario. The lithology of the roof is affected by changes in ground pressure. since the mesh approach was applied to the impacted area This type of effective solution implemented here demonstrates that this strategy contributes to safe mining. We believe that using numerous working faces with varied roof lithology is ineffective. As a result, the equipment performs poorly. To achieve a better working face, we need always pay more attention to our equipment and machinery. This research provided important information regarding working face parameters and roof control design in the same mining area with varying roof conditions. This was important due to the significant variances in coal seam characteristics and roof differences. This study emphasized the significance of The selection of fully mechanized mining equipment and ground pressure control was based on the geological evaluation of surrounding rock and the overall prediction of ground pressure within the mining region. As you read more in this work, you will notice the use of ground pressure theory in conjunction with experimental data and fault occurrence analysis. This was significant because we gained a better understanding of our working face.

Author Response

Thank you very much for your warm work. The response to your comments can be found in the attachment.

Reviewer 2 Report

Reviewer Comments

Paper title: Characteristics of strata behavior and differentiated control of fully mechanized mining working face with abnormal roof

The present manuscript based on experimental analysis and theoretical calculations, was ana-lyzed the characteristics of ground pressure in fully mechanized mining working face of the zone with abnormal roof and the action mechanism of roof lithology change on ground pressure. In the paper are proposed the mechanisms for selecting proper fully mechanized mining equipment and ground pressure control methods. Received data is of great significance for a safe and efficient mining. A manuscript has a practical application and also provides important theoretical for the next studies.

The paper can be accepted for publication after providing the corrections mentioned below.

Recommendation 1. In the Introduction section, an enhanced literature review is required. For this study, the authors have used only 21 peer-reviewed reference sources. It seems insufficient for such type of research. It will be great if the authors show some description in context – Why it is important to conduct this study?

Question 1. Can the expected result be used or implemented within other coal basins or geological conditions? If yes, then how? What limitations?

Recommendation 2. The aim and the tasks must be highlighted at the end of the Introduction section.

Question 2. Maybe it will be better to use Case study instead of Engineering overview in the title of the section 2?

Recommendation 3. You should indicate in the first sentence of the section 2 the country. “The A3 coal was obtained from group A coal roof in layers of Panbei Mine of Huainan mining area, China”.

Recommendation 4. Please add a full text description of the parameters mentioned in the Table 1 (q1-q3).

Question 3. What software were used to provide a numerical simulation and Why?

Question 4. What software were used to interpreted results of abutment pressure described on the Figure 8?

Recommendation 5. Why do not compared research results with the previous one (already known)?

Recommendation 6. Please provide a short description of further research at the end of the discussion section.

Recommendation 7. The novelty of the paper must be highlighted in the conclusions section.

Recommendation 8. There are papers that were reviewed by the reviewer in the past years. Please consider the suggested research in your paper when enhancing the literature review*. I believe they are worth considering in your paper.

Dychkovskyi, R., Shavarskyi, Ia., Saik, P., Lozynskyi, V., Falshtynskyi, V., & Cabana, E. (2020). Research into stress-strain state of the rock mass condition in the process of the operation of double-unit longwalls. Mining of Mineral Deposits, 14(2), 85-94. https://doi.org/10.33271/mining14.02.085

Vu, T.T (2022). Solutions to prevent face spall and roof falling in fully mechanized longwall at underground mines, Vietnam. Mining of Mineral Deposits, 16(1), 127-134. https://doi.org/10.33271/mining16.01.127

Wang, J., Yu, B., Kang, H., Wang, G., Mao, D., Liang, Y., & Jiang, P. (2015). Key technologies and equipment for a fully mechanized topcoal caving operation with a large mining height at ultra-thick coal seams. International Journal of Coal Science & Technology, 2(2), 97-161.

Babets, D., Sdvyzhkova, O., Shashenko, O., Kravchenko, K., & Cabana, E.C. (2019). Implementation of probabilistic approach to rock mass strength estimation while excavating through fault zones. Mining of Mineral Deposits, 13(4), 72-83. https://doi.org/10.33271/mining13.04.072

Final decision. In general, I must admit that a very good study was performed, and I will recommend your paper for publication after careful revision.

Author Response

(The authors gave the same response as above.)

Reviewer 3 Report

Lithology and structural characteristics of coal seam roof have a strong impact on the design parameters of fully mechanized mining working face, equipment selection, and ground pressure control. The article is interesting and important.  But the article still contains several shortcomings:

l  Omit the unnecessary information from the abstract section and add only key information.

l  Discuss the Novelty and clear application of the work.

l  However, little research has been performed to...Pls consider revising the statement at it different from the topic and what is done in this work.

l  In the Introduction section, an enhanced literature review is required. For this study, the authors have used only 22 reference sources. It seems insufficient for such type of research. It will be great if the authors show some description in context – Why it is important to conduct this study?

l  How were the parameters evaluated in Table 1?

l  Add the reference for each equation.

l  Add the boundary condition and complete detail about the modelling part.

l  The methodology is a bit weak, and the methodology must be improved.

l  Comparison with the previously achieved results are welcome.

l  The conclusion is superficial and needs to reflect the findings of the research in greater detail.

l  In present it looks like a technical report not a paper. Revise the manuscript carefully.

l  There are papers that I have reviewed in the past years. Please consider the suggested research in your paper when enhancing the literature review (Be aware that there are no references that belong to the reviewer). I believe they are worth considering in your paper.

1.       Li G, Ma FS, Guo J, et al. 2020. Study on deformation failure mechanism and support technology of deep soft rock roadway, Engineering Geology, 264, 105262. https://doi. org/10.1016/j.enggeo.2019.105262

2.       Sun ZY, Zhang DL, Fang Q, et al. Analysis of the interaction between tunnel support and surrounding rock considering pre-reinforcement. Tunnelling and Underground Space Technology, 2021, 115: 104074.

3.       Bahrani N, Hadjigeorgiou J. Influence of stope excavation on drift convergence and support behavior: insights from 3d continuum and discontinuum models. Rock Mechanics and Rock Engineering, 2018, 51(8): 2395–2413. https://doi.org/10.1007/s00603-018-1482-5

Author Response

(The authors gave the same response as above.)

Reviewer 4 Report

To be honest, this paper does not present any new things. The "theory" the authors used is very conventional, and the proposed protection measurements are also very common in the mining area (I believe every miner know these!). I strongly suggest the authors to imporve the novelty of the paper. 

Author Response

(The authors gave the same response as above.)

Round 2

Reviewer 2 Report

Dear authors,

I am more than satisfied with the corrections provided by you.

This study is an important contribution to sustainable mining.

Congratulations to the authors.

Author Response

Thank you for your comments. We worked with native English speaker to improve the manuscript and made some changes and rearranged the language appropriately in the revised manuscript. And here we did not list the changes, but the color of text changed will become red in the revised manuscript. We hope that the correction will meet with approval, and we are also willing to further improve the English if it is required or necessary.

Reviewer 3 Report

The authors have addressed almost all the points raised by reviewers.

Author Response

(The authors gave the same response as above.)

Reviewer 4 Report

In my opinion, the novelty of the paper still needs to be improved. I also have the following comments: 

1) Please improve the quality of the figures, for example, figure 1 is not clear enough; Figures 7 and 8 are not clear when printed in black and white. 

2) The mechanical and physical parameters of the materials in the numerical models are very important. Please state clearly that how these parameters are determined. How do you think the results are reliable? Why the strength parameters cohesion and internal friction angle after blasting are 0.5 times that before blasting? 

3) English language of the paper has to be improved. I can glance some errors when reading the paper. 

4) Line 365: what is key layer? how do you find it? In the studied case, where is the key layer? Can you present the result? 

Round 3

Reviewer 4 Report

Current version is fine for publication, although the novelty of the paper is still not high.